# Orally administered Chinese herbal therapy to assist post-surgical recovery for chronic rhinosinusitis—A systematic review and meta-analysis

Jing Cui[1], Wenmin Lin[2,3], Brian H. May[1], Qiulan Luo[2,3], Christopher Worsnop[4], Anthony Lin Zhang[1], Xinfeng Guo[2], Chuanjian Lu[2], Yunying Li[2,3], Charlie C. Xue[1,2]*

1 China-Australia International Research Centre for Chinese Medicine, School of Health and Biomedical Sciences, RMIT University, Bundoora, Victoria, Australia, 2 The Second Affiliated Hospital of Guangzhou University of Chinese Medicine, Guangdong Provincial Hospital of Chinese Medicine, Guangdong Provincial Academy of Chinese Medical Sciences, and The Second Clinical College, Guangzhou University of Chinese Medicine, Guangzhou, China, 3 Otorhinolaryngology Head and Neck Department, Guangdong Provincial Hospital of Chinese Medicine, Guangzhou, China, 4 Department of Respiratory Medicine, Austin Health, Heidelberg, Victoria, Australia

* charlie.xue@rmit.edu.au

**Data Availability Statement:** All relevant data are within the manuscript and its Supporting Information files.

## Abstract

This systematic review and meta-analysis aims to: assess the effectiveness and safety of orally administered Chinese herbal medicines (CHMs) as adjuncts to the post-surgical management of chronic rhinosinusitis (CRS); inform clinicians of the current evidence; identify the best available evidence; and suggest directions for further research. Randomised controlled trials (RCTs) were identified from searches of nine databases plus clinical trial registries. Participants were adults and/or children diagnosed with sinusitis or rhinosinusitis, with or without nasal polyps, who had received surgery. Interventions were CHMs used orally following surgery for CRS as additions to conventional post-surgical management. Controls received conventional post-surgical management without CHMs. Studies reported results for Sino-Nasal Outcome Test (SNOT), visual analogue scales (VAS), Lund-Mackay computed tomography score (LM), Lund-Kennedy endoscopic score (LK), mucociliary transport time (MTT), mucociliary transport rate (MTR), mucociliary clearance (MC) or quality of life (QoL). Twenty-one RCTs were included. All used oral CHMs following functional endoscopic sinus surgery (FESS). The pooled results showed no significant difference between groups for SNOT-20 at the end of treatment (EoT) but there was a significant difference at follow up (FU) in favour of additional CHMs. The VAS for total nasal symptoms (VAS-TNS) showed greater improvements in the CHM groups at EoT and FU. Only FU data were reported for LM which showed greater improvement in the CHM groups. LK showed greater improvements at EoT and FU. The measures of mucociliary transport (MTT, MTR, and MC) each showed significantly greater improvement at EoT in the group that received additional CHMs. No study reported QoL. Adverse events were not serious, but reporting was incomplete. The meta-analyses suggested the addition of oral CHMs to conventional management following FESS may improve recovery. However, most studies were not blinded, and

**Funding:** Funding support was provided by the China-Australia International Research Centre for Chinese Medicine (CAIRCCM) - a joint initiative of RMIT University, Australia and Guangdong Provincial Academy of Chinese Medical Sciences, China. The funders had no role in study design, data collection and analysis, decision to publish, or preparation of the manuscript.

**Competing interests:** The authors have declared that no competing interests exist.

substantial heterogeneity was evident in some meta-analyses. Blinded studies are required to further investigate the roles of oral CHMs in post-surgical recovery.

**Systematic review registration number:** The protocol was registered in PROSPERO (CRD42019119586).

## Introduction

Rhinosinusitis is an inflammation of the paranasal sinuses and nasal cavity with the symptoms of: nasal blockage, obstruction and/or congestion; or nasal discharge (anterior / posterior nasal drip), with or without facial pain/pressure and/or reduction or loss of smell, with verification using endoscopy or computed tomography (CT) [1–3]. When two or more symptoms, including nasal blockage, obstruction and/or congestion or nasal discharge, persist for 12 weeks or more it is chronic rhinosinusitis (CRS). People with CRS can be broadly divided into those with nasal polyps (CRSwNP) and those without nasal polyps (CRSsNP) [2, 3].

The prevalence of CRS has been estimated at 10.9% of the European adult population based on surveys, but there was wide variation between countries [3, 4]. In the American population, the estimate was 12.1% [3, 5]. In eastern Asia, CRS prevalence was 8.0% (4.8–9.7%) in a survey of seven Chinese cities [6], and in South Korea it was 7.0% based on a survey plus physical examination [7], and 10.8% in a symptom-based survey [8]. In other surveys, when imaging was used in addition to symptoms, the prevalence of CRS was markedly lower [3]. Estimates of the prevalence CRSwNP were 2.7% in Sweden [3, 9], 2.11% in France [3, 10] and 2.5% in Korea [3, 11].

The conventional pharmacological management of CRS usually involves intra-nasal steroids, nasal saline irrigation, and/or oral antibiotic therapy during exacerbations, with allergen avoidance in people with allergies [2, 3]. Biologics have recently been introduced [12].

Various complementary therapies have been used for rhinosinusitis, including herbal medicines [13, 14], homeopathy [15–17], and acupuncture therapies [18–21]. Nevertheless, when these approaches do not provide sufficient control of symptoms, such as facial pain and headache [22], surgery can be effective [1, 3, 23, 24].

In the United States, more than 250,000 people undergo endoscopic sinus surgery (ESS) for CRS each year [25]. In Germany, 17.33% of people had received functional endoscopic sinus surgery (FESS) within 12 months of diagnosis of CRSwNP [26]. Improvements in quality of life [27], symptoms, and productivity [28] and medication use [29] following FESS have been reported. However, a 12-year follow-up study in Belgium reported that although there were still improvements in symptoms, 78.9% of people showed recurrence of nasal polyps [30]. Another observational survey reported that following FESS 17% of participants had received revision surgery with 10 years [31].

In China the main indications for surgery for CRS are variation in the anatomical structure of the airway blocking mucous drainage, presence of nasal polyps, poor response to medical treatment after 12 weeks, and complications of CRS [32]. International guidelines have proposed that for people with uncomplicated CRS refractory to medical therapy for at least eight weeks and, following a CT scan that showed an LM score ≥ 1, surgery may be offered [3, 33, 34]. Following FESS, comprehensive treatment is the key to ensure epithelialisation of the operative cavity mucosa and control of symptoms. Comprehensive treatment includes nasal or oral glucocorticosteroids, nasal irrigation, management of the operative sinus cavity, and management of any other issues [35].

Additional managements for CRS can include herbal medicines [36]. In China, Japan and Korea, traditional herbal medicines have roles within the state-based health care systems [37–40]. In 2021, there were 4,630 Chinese medicine hospitals and 756 integrative Chinese–Western medicine hospitals in mainland China [41]. The Chinese guidelines for the diagnosis and treatment of CRS have included Chinese herbal medicines (CHMs) based on syndrome differentiation as additional treatment options [32, 42]. The CHMs can be administered orally and/or nasally [43, 44].

A review of various types of herbal medicines for CRS reported improvements in symptoms [36]. A systematic review of 34 studies of oral and/or nasal CHMs for acute or chronic rhinosinusitis, without surgery, found improvements in symptoms, sinus imaging and measures of mucociliary clearance [43]. Another systematic review that included 80 randomised controlled trials (RCTs) of herbal medicines for CRS, mostly from China, reported improvements associated with the herbal medicines, but that study combined studies of CRS without surgery and CRS post-surgery [45]. A review of 32 RCTs of CHMs following surgery reported benefits for combining CHM nasal irrigations with conventional medications [46]. We were unable to locate a systematic review that evaluated the effects of orally administered herbal medicines following FESS for CRS.

The objectives of this systematic review were to: assess the effects and safety of CHMs in the post-surgical management of people with CRS; locate the best available evidence; inform clinicians of the current state of the evidence; and identify directions for further research.

## Materials and methods

This review was in accord with the PRISMA guidelines [47, 48], the extension for CHM [49], and Cochrane Collaboration methodology [50, 51]. The systematic review protocol was registered with PROSPERO (CRD42019119586).

### Selection criteria

Prospective RCTs were included with no limitations on language or publication type. Retrospective studies were excluded.

**Participants.**   Were adults and/or children diagnosed with sinusitis or rhinosinusitis with or without nasal polyps who had received surgery for their condition. Studies were excluded when any participants had fungal RS or non-RS nasal conditions, the condition was RS but not during post-surgical recovery, and/or the interventions included nasal CHMs.

**Test interventions.**   Were orally administered CHMs used in eastern Asia (China, Korea, Japan). Forms could include liquids, granules, capsules or pills. Injections and single chemical compounds were excluded. The CHMs were administered following surgery for CRS as an integrative medicine (IM) intervention. The addition of CHMs pre-surgery was allowed. Conventional post-surgical management was applied in both groups.

**Control interventions.**   Included conventional post-surgical management that was the same as in the test group. This could include antibiotics, other pharmacotherapies (oral and/or nasal), nasal irrigations, sprays, and/or inhalations, as in guidelines [1–3, 32, 52–56]. The addition of a placebo for the CHM was allowed.

**Outcome measures.**   Were Sino-Nasal Outcome Test (SNOT), including SNOT20 [57], SNOT22 [58] and SNOT8 [59], visual analogue scale for total nasal symptoms (VAS-TNS) and/or VAS for individual symptoms (VAS-IS), Lund-Mackay computed tomography score (LM) [60], Lund-Kennedy endoscopic score (LK) [61, 62], Mucociliary transport time (MTT) [63], Mucociliary transport rate (MTR), Mucociliary clearance as percentage (MC), and

adverse events (AEs) [64]. Effective rate scales not used internationally, and measures developed by the authors were excluded.

**Settings.** Included inpatients and outpatients following discharge from hospital.

### Information sources and search strategy

Five English-language and four Chinese-language databases were searched from their respective inception dates until 9 August 2022, with no limit on year. Searches of four clinical trial registries, Web of Science, and ProQuest Central were conducted on 12 August 2022 with no limits. Reference lists in retrieved papers were also searched. The information sources and search terms are listed in S1 File.

### Data extraction

Study characteristics and outcome data were extracted to predefined spreadsheets by JC and WML and independently checked by JC, BHM and QL. Any issues were resolved by discussion between reviewers with ALZ as final arbiter.

### Risk of bias assessment

Risk of bias was assessed independently by two reviewers (JC, BHM) and mediated by a third (ALZ) using the Cochrane tool [50] for sequence generation, allocation concealment, blinding of participants, blinding of personnel, blinding of outcome assessment, incomplete outcome data, and selective outcome reporting. Potential publication bias was assessed using Funnel plots when 10 or more studies were available.

### Data analysis

Analysis was conducted in Review Manager (version 5.4.1). Mean difference (MD), risk ratio (RR) and 95% confidence intervals (CI) were calculated, and heterogeneity was measured using the $I^2$ statistic. Conservative random-effect models were used due to likely heterogeneity in study populations and methods. Baseline scores were compared between groups to determine baseline comparability. Planned sensitivity analyses explored any effects of baseline imbalances, study duration, and/or herbal formula. Post-hoc sensitivity analyses included presence or absence of nasal polyps and Chinese medicine syndrome. The certainty of the evidence was assessed using Grading of Recommendations Assessment, Development and Evaluation (GRADE) [65, 66].

## Results

### Literature search results

Search results were downloaded to separate spreadsheets and combined for screening. After removal of duplications, 8,785 records remained. Based on titles, abstracts and other information, 8,490 records were excluded, and 295 full text papers relating to CHM were obtained for further assessment against the inclusion and exclusion criteria. Twenty-one RCTs satisfied the selection criteria (Fig 1). These enrolled 1,997 participants. All studies were conducted in China. All studies used FESS and some mentioned use of the Messerklinger surgical method [32, 67, 68].

Three studies included adolescents [69–71]. The ages of participants ranged from 15–74 years and CHM treatment durations ranged from 2–12 weeks. In seven studies all participants had nasal polyps (CRSwNP); in six studies all were without polyps (CRSsNP). The remaining

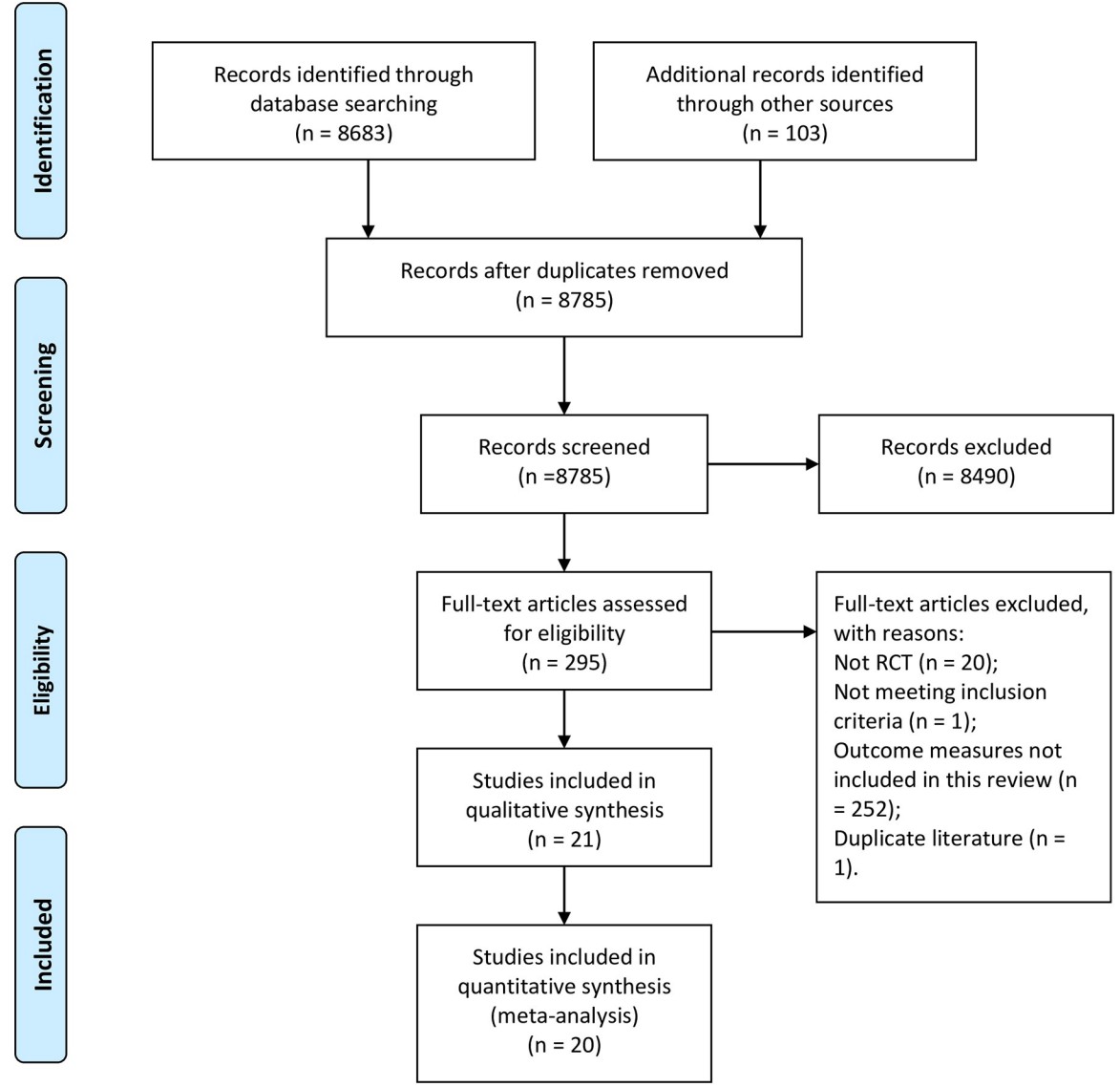

**Fig 1. Flow diagram of the search and study selection process.** Abbreviations: RCT: randomized controlled trial.

eight studies were of mixed groups (Table 1). One study included three groups [72]. One study employed a placebo for the CHM in the control group [73].

## Interventions

The 21 RCTs investigated 13 different CHM formulae. The most frequent was *Bi yuan tong qiao ke li* (BYTQKL) (3 studies). Two formulae, *Huang qin hua shi tang* (HQHST) and *Shen ling bai zhu san* (SLBZS), were used in two studies each. The following 10 formulae were each used in one study: *Qing re li shi qu yu tang* (QRLSQYT), *Xin qian gan ju tang* (XQGJT), *Jian pi hua zhuo tong qiao fang* (JPHZTQF), *Bi yan kang pian* (BYKP), *Huo dan wan* (HDW), *Qing bi tang* (QBT), *Shu du tang* (SDT), *Wen fei zhi liu dan* (WFZLD), *Wu wei xiao du yin* (WWXDY) and *Bi yan ning ke li* (BYNKL). Three studies used unnamed CHM formulae (Table 1) [70, 74, 75].

**Table 1. Characteristics of included studies.**

| Study ID; duration | No. participants enrolled (T,C); no. with polyps; age; gender (M/F); no. completed (T,C) | Interventions | | Outcome measures included in this review |
|---|---|---|---|---|
| | | CHM; dosage; syndrome | Post-surgical care used in T and C | |
| Chen WM 2013; 3 mths | 50 (25,25); 50; 20–55 y; (31/19); (25,25) | *Sheng ling bai zhu san* (SLBZS) plus PT; one packet per day, twice daily; NS. | Oral macrolides, oral expectorant, nasal wash (not specified) | VAS-TNS; MTR |
| Fan ZJ 2013; 12 wks | 82 (41,41); 53; 18–60 y; (NS); (41,41) | Unnamed CHM formula plus PT; one packet per day; NS. | Routine post-surgical treatment, oral clarithromycin, fluticasone propionate nasal spray[a] | MTR |
| Fu SW 2015; 2 mths | 90 (45,45); 0; 16–67 y; (61/29); (45,45) | *Huo dan wan* (HDW) plus PT; 6 g per dose, 3 times daily; NS. | Routine antibiotics and oral corticosteroids | SNOT-20; LK |
| Gao Y 2022; 3 mths | 237 (79,79,79); 237; 20–60 y; (125/112); (79,79,79) | 1. *Bi yan ning ke li* (BYNKL) plus PT without budesonide nasal spray; 15 g per dose, 3 times daily; NS. 2. *Bi yan ning ke li* (BYNKL) plus PT; 15 g per dose, 3 times daily; NS. | Routine antibiotics with or without budesonide nasal spray | SNOT-20-IS; LK-IS; VAS-IS |
| Li YX 2014; 4 wks | 63 (NS); 60; 18–74 y; (36/27); (30,30) | Unnamed CHM formula plus PT; decoction, twice daily; Cold and damp coagulation & damp-heat steaming. | Oral macrolides, budesonide nasal spray | SNOT-20; SNOT-20-5i; LK; VAS-IS |
| Liu H 2016; 8 wks | 64 (32,32); 0; 20–66 y; (33/31); (32,32) | *Jian pi hua zhuo tong qiao fang* (JPHZTQF) plus PT; one packet per day, twice daily; NS. | The post-surgical treatment was not described in the article. | MTT; MTR; MC |
| Mou S 2015; 4 wks | 40 (20,20); 40; 18–65 y; (21/19); (20,20) | *Shu du tang* (SDT) from 4 weeks after surgery plus PT; 100 mL per dose, twice daily; Stagnant heat in the Gallbladder. | Routine antibiotic treatment after surgery plus oral placebo for CHM from 4 weeks after surgery. | VAS-TNS; LK |
| Shu YY 2013; 2 wks | 60 (30,30); NS[b]; 19–68 y; (39/21); (30,30) | *Wu wei xiao du yin* (WWXDY) plus PT; 250 mL per dose, twice daily; Heat accumulation in the Lung meridian. | Cefuroxime sodium intravenous drip, oral Myrtol® standardized enteric coated soft capsules, oral cefuroxime axetil tablets, budesonide nasal spray | VAS-IS; LK |
| Tan GD 2011; 3 mths | 100 (50,50); 100; 18–67 y; (54/46); (50,50) | Unnamed CHM formula plus PT; one packet per day, twice daily; Lung and Spleen *qi* deficiency. | Routine antibiotic intravenous drip, glucocorticoids and haemostatics intravenous drip, oral antihistamines, budesonide nasal spray | MTR |
| Wang Y 2017; 4 wks | 80 (40,40); 0; 42.9 ± 4.5 y; (49/31); (40,40) | *Bi yuan tong qiao ke li* (BYTQKL) plus PT; 15 g per dose, 3 times daily; Stagnant heat in the Gallbladder. | Cephalosporin intravenous drip, dexamethasone injection, oral cefradine capsules | VAS-IS (pain); MTT; MTR; MC |
| Wang YJ 2015; 2 wks | 150 (75,75); 108; 19–67 y; (78/72); (75,75) | *Bi yuan tong qiao ke li* (BYTQKL) plus PT; 15 g per dose, 3 times daily; NS. | Cephalosporin plus dexamethasone intravenous drip, oral antibiotics (not specified) | MTT; MTR; MC |
| Zeng YS 2016; 6 wks | 72 (36,36); 0; 20–70 y; (40/32); (36,36) | *Huang qin hua shi tang* (HQHST) plus PT; one packet per day, twice daily; Spleen-Stomach dampness heat. | Antibiotics by intravenous drip, budesonide nasal spray | LK; LM; VAS-TNS |
| Zeng YS 2021; 6 wks | 60 (30,30); 60; 25–68 y; (38/22); (30,30) | *Huang qin hua shi tang* (HQHST) plus PT; one packet per day, twice daily; Spleen-Stomach dampness heat. | Antibiotics by intravenous drip, budesonide nasal spray | LK; VAS-TNS; MTR |
| Zhang EQ 2018; 15 d | 102 (51,51); NS[b]; 15–65 yrs (72/30); (51,51) | Unnamed CHM formula plus PT; decoction, twice daily; NS. | Routine post-surgical care, mometasone furoate nasal spray | SNOT-20; LK |
| Zhang J 2016; 4 wks | 100 (50,50); 0; 18–60 y; (54/46); (50,50) | *Bi yuan tong qiao ke li* (BYTQKL) plus PT; 15 g per dose, 3 times daily; Stagnant heat in the Gallbladder. | Cephalosporine and dexamethasone intravenous drip, oral cefradine capsules | VAS-IS (pain); MTT; MTR; MC |
| Zhang JY 2019; 12 wks | 166 (83,83); NS[b] 18–60 y; (109/55); (82,82) | *Xin qian gan ju tang* (XQGJT) plus PT; 150 mL per dose, twice daily; NS. | Routine post-surgical care, mometasone furoate nasal spray (Nasonex) | LK; LM; VAS-TNS |
| Zhang R 2021; 12 wks | 92 (46,46); 60; 23–69 y; (53/39); (46,46) | *Qing re li shi qu yu tang* (QRLSQYT) plus PT; 200 mL per dose, twice daily; Spleen-Stomach dampness heat. | Etamsylate intravenous drip, penicillin-type intravenous drip, budesonide nasal spray | MTT; MTR; MC |
| Zhang WQ 2020; 4 wks | 60 (30,30); 60; 18–65 y; (29/31); (30,30) | *Shen ling bai zhu san* (SLBZS) plus PT; 250 mL per dose, twice daily; Spleen *qi* deficiency. | Routine antibiotic treatment, oral eucalyptol, limonene and pinene enteric soft capsules, mometasone furoate nasal spray | LK; VAS-TNS |
| Zheng XR 2017; 2 mths | 78 (39,39); 0; 20–58 y; (36/42); (39,39) | *Bi yan kang pian* (BYKP) plus PT; 3 tablets per dose, 3 times daily; NS. | Routine antibiotic treatment after surgery | VAS-TNS |

*(Continued)*

**Table 1.** (Continued)

| Study ID; duration | No. participants enrolled (T,C); no. with polyps; age; gender (M/F); no. completed (T,C) | Interventions | | Outcome measures included in this review |
|---|---|---|---|---|
| | | CHM; dosage; syndrome | Post-surgical care used in T and C | |
| Zhou ML 2016; 4 wks | 60 (30,30); NS[b] 18 65 y; (41/19); (30,30) | *Wen fei zhi liu dan* (WFZLD) plus PT; one packet per day, twice daily; Lung *qi* deficiency with cold. | Antibiotic intravenous drip, oral eucalyptol, limonene and pinene enteric soft capsules, budesonide nasal spray | LK; VAS-TNS |
| Zhu HH 2014; 8 wks | 194 (96,98); 131; 16–65 y; (108/86); (96,98) | *Qing bi tang* (QBT) plus PT; one packet per day, twice daily; NS. | Antibiotic intravenous drip, oral amoxicillin, gentamicin, dexamethasone nasal wash | LK; MTR |

Abbreviations: C: control group; CHM: Chinese herbal medicine; d: days; LK: Lund-Kennedy Endoscopic score; LK-IS: Lund-Kennedy Endoscopic individual symptoms; LM: Lund-Mackay computed tomography (CT) score; MC: mucociliary clearance; MTR: mucociliary transport rate; MTT: mucociliary transport time; mths: months; No: number; NS: not specified; PT: pharmacotherapies; Sino-Nasal Outcome Test (SNOT)-20; SNOT-20-5i: Sino-Nasal Outcome Test the top five items; SNOT-20-IS: Sino-Nasal Outcome Test individual symptoms; T: test group; VAS-IS: Visual Analog Scale—scores for individual symptoms; VAS-TNS: Visual Analog Scale—scores for total nasal symptoms; wks: weeks; y: years.

[a]The description of the post-surgical management was unclear and appeared to indicate that oral clarithromycin was only used in the control group; however, this appears to be an error, so we have assumed that both groups received oral clarithromycin plus fluticasone propionate nasal spray, since this was common practice.

[b]The participants could have nasal polyps or no polyps; numbers were not provided for these groups.

Chinese medicine (CM) syndrome differentiation was used in 11 studies. The most frequent syndromes were Spleen–Stomach dampness heat (*pi wei shi re*) (*n* = 3) and Stagnant heat in the Gallbladder (*dan fu yu re*) (*n* = 3). Further details on the test interventions, manufacturing, ingredients, diagnostic criteria, and sources of study funding are in S1 File.

Many CHMs shared common ingredients (S1 File). The most frequent were *Magnolia biondii* Pamp. (*xin yi*) (*n* = 15), *Angelica dahurica* (Fisch. ex Hoffm.) Benth. et Hook. f (*bai zhi*) (*n* = 14), *Glycyrrhiza uralensis* Fisch (*gan cao*) (*n* = 13), *Xanthium sibiricum* Patr. (*cang er zi*) (*n* = 12), *Scutellaria baicalensis* Georgi (*huang qin*) (*n* = 11), *Poria cocos* (Schw.) Wolf (*fu ling*) (*n* = 7), and *Astragalus membranaceus* (Fisch.) Bge. var. *mongholicus* (Bge.) Hsiao (*huang qi*) (*n* = 7). Each of these herbs was a common ingredient in CHM formulae for CRS [43, 76]. This suggests that the herbal interventions tested in the RCTs were broadly representative of CHM formulae for CRS. In the subgroup of seven studies that only included participants with CRCwNP (S1 File), the frequencies of the ingredients were very similar to the totals (S1 File). However, in the subgroup of six studies of CRCsNP, the most frequent herb was *Scutellaria baicalensis* Georgi (*huang qin*) (*n* = 4).

### Risk of bias

Ten studies were judged low risk for sequence generation [71–73, 77–83]. All studies were judged 'unclear' for allocation concealment because specific information was not provided. One study [73] was judged unclear risk for blinding of participants because a placebo for the CHM was used in the control group, but there was no description of whether participants were blinded to group allocation. It was judged as high risk for blinding of personnel and blinding of outcome assessment due to lack of information. The remaining studies were each judged high risk for blinding of participants, personnel and outcome assessors [69–72, 74, 75, 77–90]. All studies were judged low risk for incomplete outcome data because there were few dropouts. All were judged unclear risk for selective outcome reporting because study protocols were unavailable (S1 File).

### Outcomes

Outcomes were between-group scores at end of treatment (EoT) and/or end of follow up (FU). Each of the studies reported one or more of the following outcomes.

**Sino-nasal outcome test.** Three studies reported SNOT-20 total score [69, 70, 74], but one study of CRSsNP only reported the result at the follow up [69]. One study of CRSwNP also reported the top five items for SNOT-20 at EoT and found that scores reduced in both groups, with a greater reduction in the CHM group (MD −0.98 [−1.93, −0.03]) (S1 File) [74]. For SNOT-20 total score at EoT, one study in a mixed group [70] found a significant difference between groups (MD −6.50 [−8.01, −4.99]), but the other study of CRSwNP showed no difference (MD −1.11 [−3.04, 0.82]) [74] (Fig 2).

In the pooled result, there were improvements in both groups but no significant difference between the two groups (MD −3.84 [−9.12, 1.44], $I^2$ = 95%, $n$ = 162), with considerable heterogeneity. At FU, one study reported multiple timepoints from 6–12 months post-surgery for the five top items and the total score [74], some of which showed greater improvements in the CHM group. The study that only reported data at the six-month FU [69] reported a significantly greater reduction in total score in the CHM group (MD −3.40 [−4.85, −1.95]). When the total scores were pooled at the longest FU (6–24 months), there was no significant difference between groups (MD −1.99 [−4.66, 0.68], $I^2$ = 89%, $n$ = 133), but the heterogeneity was considerable (S1 File). In a sensitivity analysis that used the same time-point (6-month FU), the difference was significant (MD −3.39 [−4.50, −2.28], $I^2$ = 0%, $n$ = 147) without heterogeneity (Fig 2).

**Visual analogue scales.** Eight studies reported VAS-TNS at EoT [73, 80, 82–84, 88–90], and two also reported results at FU [88, 89]. In the pooled result of eight studies, there were greater reductions in VAS-TNS in the CHM groups (MD −0.89 [−1.36, −0.41], $I^2$ = 87%, $n$ = 576) with considerable heterogeneity (Fig 3).

A sensitivity analysis of five studies that reported baseline data—each of which showed no significant differences at baseline [73, 80, 82, 83, 90]—also found a significantly greater improvement in the CHM group with reduced heterogeneity (MD −1.08 [−1.72, −0.44], $I^2$ = 61%, $n$ = 404) (S1 File), but the pooled result at baseline showed the CHM groups tended to be marginally more severe (MD 0.38 [0.01, 0.75] $I^2$ = 0%, $n$ = 404). In a further sensitivity analysis of studies that used the same treatment duration (4 weeks) [73, 82, 83], there were greater improvements in both groups and a significant additional benefit in the CHM group (MD −1.11 [−1.65, −0.57], $I^2$ = 0%, $n$ = 160) with no heterogeneity. In the two studies that tested *Shen ling bai zhu san*, in which all participants had the same CM syndrome, these was a significantly greater reduction in symptom scores in the CHM group (MD −1.56 [−1.99, −1.14], $I^2$ =

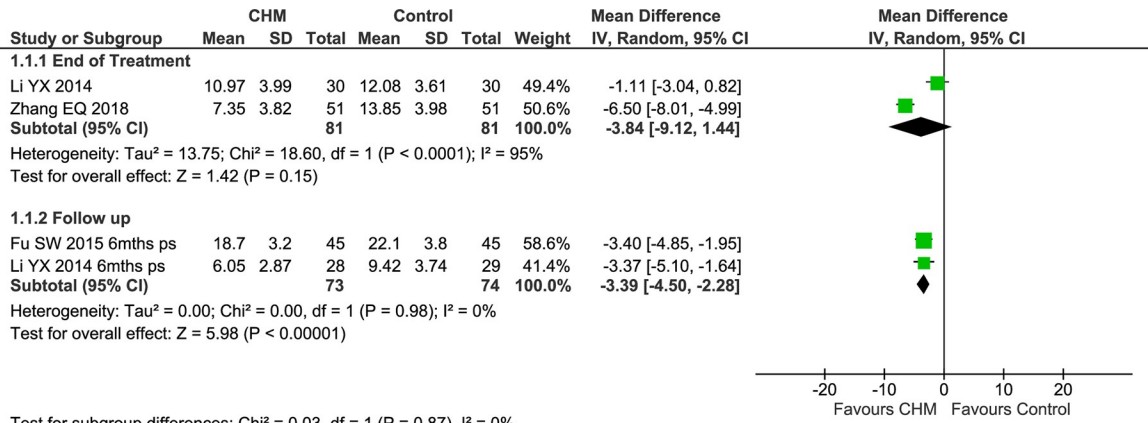

**Fig 2. Forest plot of CHM for CRS post-surgery: SNOT-20 total score.** Abbreviations: CHM: Chinese herbal medicine; SNOT-20: Sino-Nasal Outcome Test-20; ps: post-surgery; mths: months.

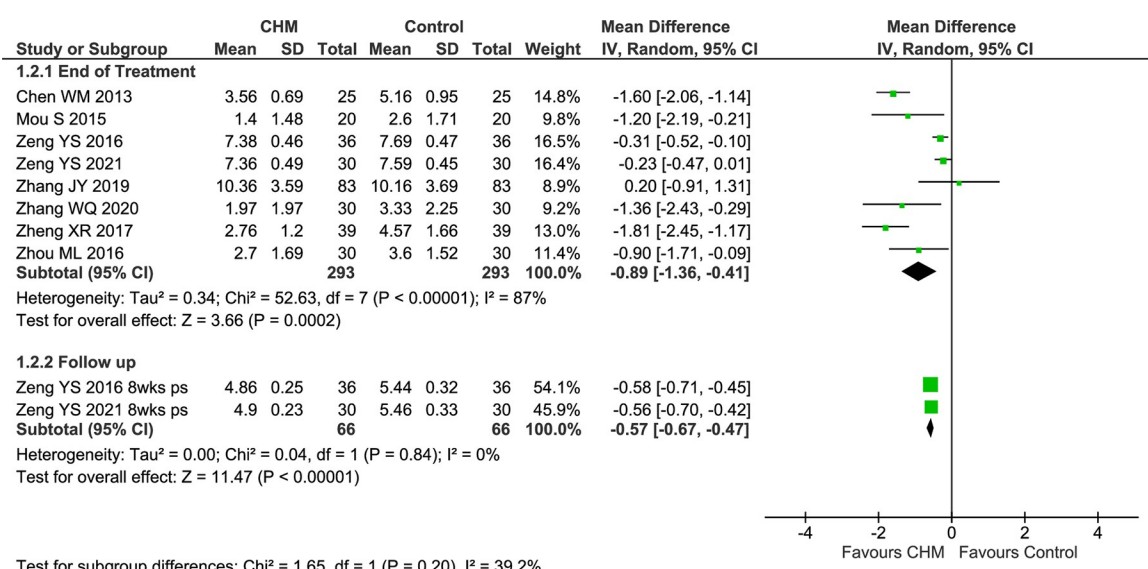

**Fig 3. Forest plot of CHM for CRS post-surgery: VAS-TNS.** Abbreviations: CHM: Chinese herbal medicine; ps: post-surgery; VAS: Visual analogue scale; TNS: total nasal symptoms; wks: weeks.

0%, $n = 110$), with no heterogeneity. A similar result was found for the two studies of *Huang qin hua shi tang* (MD −0.27 [−0.43, −0.11], $I^2 = 0\%$, $n = 120$), with no heterogeneity. In the four studies of CRSwNP, the result was significant in favour of the CHM groups (MD−1.06 [−1.95, −0.17], $I^2 = 90\%$, $n = 210$), but in pooled result for the two studies of CRSsNP there was no significant difference (MD −1.03 [−2.50, 0.44], $I^2 = 95\%$, $n = 150$), and there was considerable heterogeneity in both results (S1 File).

Two studies provided FU data at 8 weeks post-surgery, and the pooled result showed a significant reduction in VAS scores with no heterogeneity (MD −0.57 [−0.67, −0.47], $I^2 = 0\%$, $n = 132$) (Fig 3).

Two studies reported VAS-IS for four symptom categories [74, 77], and another two studies reported VAS for pain only [78, 79] (S1 File). In one study [77], dizziness and headache were combined into one category, and in one study [74], discharge showed a significant imbalance at baseline, so data were poolable for three categories: congestion (2 studies), olfactory decline (2 studies) and pain (3 studies). The pooled results at EoT showed greater reductions in congestion, olfactory decline, and pain, each with no heterogeneity. In a sensitivity analysis of two studies that used the same CHM for the same CM syndrome, there was a significant reduction in pain in the CHM group (MD −0.89 [−1.02, −0.77], $I^2 = 0\%$, $n = 180$) (S1 File).

**Lund-Mackay computed tomography score.** Two studies reported results for LM at FU only [80, 88]. In the pooled FU results of these two studies, there was a greater reduction in LM scores in the groups that received CHMs (MD −0.64 [−1.00, −0.28], $I^2 = 18\%$, $n = 228$) without important heterogeneity (Fig 4, S1 File).

**Lund-Kennedy endoscopic score.** Eleven studies reported LK [69–71, 73, 74, 77, 80, 82, 83, 88, 89], but one only reported results at 12 months post-surgery [70]. Five studies did not report baseline data (S1 File). The pooled results of 10 studies at EoT [69, 71, 73, 74, 77, 80, 82, 83, 88, 89] showed that there was a greater improvement in LK in the groups that received CHMs (MD −0.63 [−0.82, −0.44], $I^2 = 59\%$, $n = 862$) with moderate heterogeneity (Fig 5). The Funnel plot did not appear to be skewed (S1 File).

A sensitivity analysis that excluded studies that did not report baseline data also found a significantly greater improvement (MD −0.86 [−1.30, −0.43], $I^2 = 72\%$, $n = 640$) but with

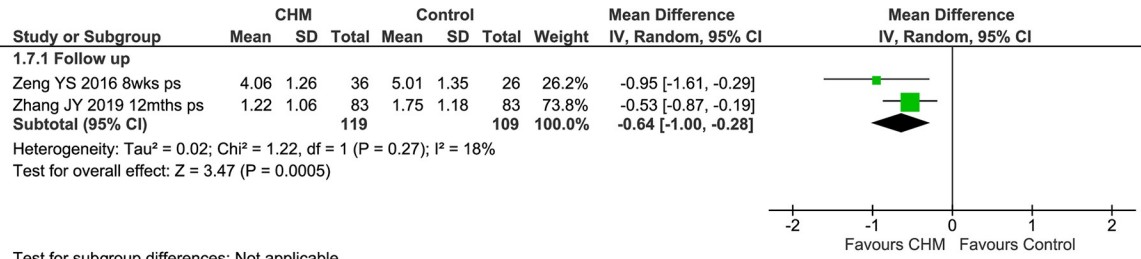

**Fig 4. Forest plot of CHM for CRS post-surgery: LM.** Abbreviations: CHM: Chinese herbal medicine; LM: Lund-Mackay computed tomography score; mths: months; ps: post-surgery; wks: weeks.

increased heterogeneity (S1 File). A further sensitivity analysis that only included four studies of the same treatment duration (4 weeks) [73, 74, 82, 83] found a greater reduction in the CHM groups (MD −0.95 [−1.38, −0.51], $I^2$ = 0%, $n$ = 220) with no heterogeneity. The two studies of *Huang qin hua shi tang*, in which all participants had the same CM syndrome, showed a significantly greater improvement in LK scores (MD −0.55 [−0.65, −0.45], $I^2$ = 0%, $n$ = 132) with no heterogeneity, but one study was of CRSwNP while the other was of CRSsNP. In the sensitivity analysis of all four studies of CRSwNP, there was a greater reduction in LK in the CHM group (MD −0.62 [−0.86, −0.39], $I^2$ = 11%, $n$ = 220), and a similar result was found for the two studies of CRSsNP (MD −0.55 [−0.68, −0.42], $I^2$ = 0%, $n$ = 162), both with reduced heterogeneity.

Six studies provided results at FU. The pooled result at the longest FUs (8 weeks – 24 months) showed significantly greater improvements in the CHM groups (MD −0.95 [−1.47, −0.42], $I^2$ = 97%, $n$ = 531) but with considerable heterogeneity (Fig 5). A sensitivity analysis that excluded three studies that did not provide baseline data found a similar result (MD −1.62

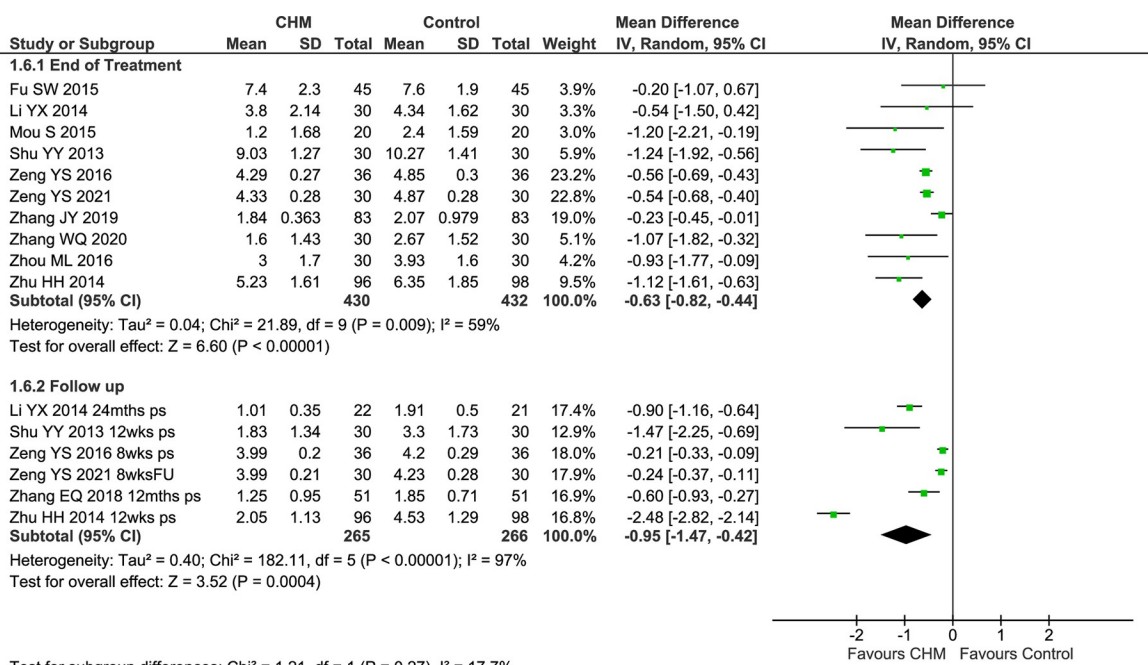

**Fig 5. Forest plot of CHM for CRS post-surgery: LK.** Abbreviations: CHM: Chinese herbal medicine; FU: follow-up; LK: Lund-Kennedy endoscopic score; mths: months; ps: post-surgery; wks: weeks.

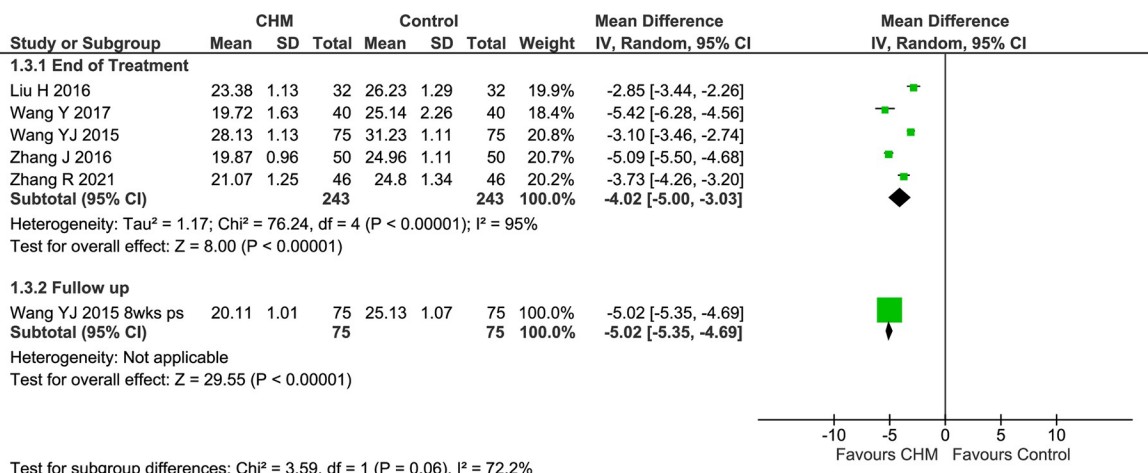

**Fig 6. Forest plot of CHM for CRS post-surgery: MTT.** Abbreviations: CHM: Chinese herbal medicine; MTT: Mucociliary transport time; ps: post-surgery; wks: weeks.

[−2.78, −0.46], $I^2$ = 96%, *n* = 314). When the FU duration was the same (12 weeks), there was still a significant benefit in favour of the CHM group (MD −2.04 [−3.02, −1.06], $I^2$ = 81%, *n* = 254), but the heterogeneity remained considerable (S1 File).

**Mucociliary transport time.** Five studies reported MTT [78, 79, 81, 86, 87], and each showed no significant baseline imbalance (S1 File). The pooled results of all five studies at EoT showed greater improvement in MTT in the CHM groups (MD −4.02 [−5.00, −3.03], $I^2$ = 95%, *n* = 486), with considerable heterogeneity (Fig 6).

A sensitivity analysis of two 4-week studies [78, 79] found significantly greater improvement in the CHM group (MD −5.15 [−5.52, −4.78], $I^2$ = 0%, *n* = 180), with no heterogeneity (S1 File). In the pooled result for the three studies of *Bi yuan tong qiao ke li*, there was a significantly greater reduction in MTT (MD −4.51 [−6.06, −2.96], $I^2$ = 97%, *n* = 330), but the heterogeneity was considerable. A similar result was found for the three studies of CRSsNP (MD −4.44 [−6.02, −2.87], $I^2$ = 95%, *n* = 244). When two studies that used the same CHM (*Bi yuan tong qiao ke li*) for the same CM syndrome in CRSsNP, there was a significantly greater reduction in MTT in the CHM groups (MD −5.15 [−5.52, −4.78], $I^2$ = 0%, *n* = 180), with no heterogeneity.

One study reported results at multiple FUs (4, 6 and 8 weeks) [87] and found significantly shorter MTT at 8 weeks post-surgery (MD −5.02 [−5.35, −4.69]).

**Mucociliary transport rate.** Ten studies reported MTR [71, 75, 78, 79, 81, 84–87, 89]. One did not report baseline data [84], and one only reported data at 12 months FU (S1 File) [71]. The pooled results of nine studies at EoT [75, 78, 79, 81, 84–87, 89] found that there was a greater improvement in MTR in the CHM groups (MD 1.56 [0.93, 2.20], $I^2$ = 98%, *n* = 778), with considerable heterogeneity (Fig 7).

In a sensitivity analysis that excluded the study that did not report baseline results [84], the result was similar (MD 1.51 [0.83, 2.18], $I^2$ = 99%, *n* = 728). When the duration of treatment was the same (12 weeks), the result was similar (MD 1.29 [0.80, 1.78], $I^2$ = 86%, *n* = 324) with reduced heterogeneity. However, one study lacked baseline data, so it was excluded in a further sensitivity analysis that found a similar result (MD 1.11 [0.65, 1.57], $I^2$ = 85%, *n* = 274), with reduced heterogeneity (S1 File). The pooled result of the two studies of four weeks duration found a significantly greater improvement in MTR in the CHM groups (MD 2.53 [1.90, 3.16] $I^2$ = 92%, *n* = 180), but the heterogeneity was considerable. For the three studies of *Bi yuan*

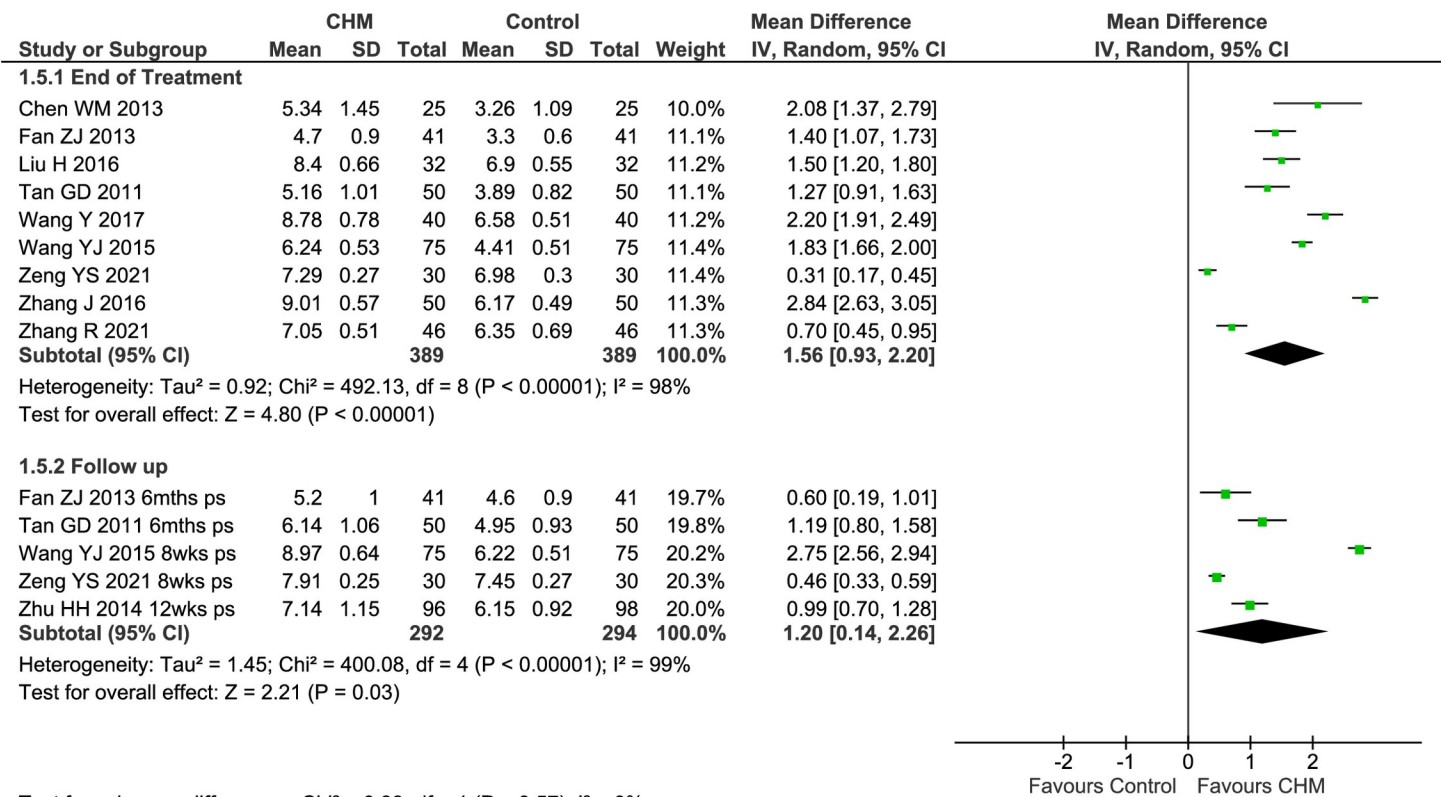

**Fig 7. Forest plot of CHM for CRS post-surgery: MTR.** Abbreviations: CHM: Chinese herbal medicine; mths: months; MTR: Mucociliary transport rate; ps: post-surgery; wks: weeks.

*tong qiao ke li*, the MTR increased significantly (MD 2.29 [1.64, 2.94], $I^2$ = 96%, *n* = 330), with considerable heterogeneity. The sensitivity analysis of three studies of CRSwNP found a significantly greater increase in the CHM groups (MD 1.17 [0.22, 2.11], $I^2$ = 95%, *n* = 210), as did the pooled result for three studies of CRSsNP (MD 2.19 [1.40, 2.97], $I^2$ = 96%, *n* = 244), but both showed considerable heterogeneity. In the two studies that tested *Bi yuan tong qiao ke li* in participants with CRSsNP and the same CM syndrome, there was a similar result (MD 2.53 [1.90, 3.16], $I^2$ = 92%, *n* = 180). When the CM syndrome was the same, but the CHM and the CRS types were different, the benefit in the pooled result of two studies was very small (MD 0.49 [0.11, 0.87], $I^2$ = 86%, *n* = 152), and the heterogeneity was considerable.

Five studies provided data at FU [71, 75, 85, 87, 89]. The pooled result at longest FU (8 weeks –6 months) showed a significantly greater increase in the CHM groups (MD 1.20 [0.14, 2.26], $I^2$ = 99%, *n* = 545), with considerable heterogeneity (Fig 7). When both studies had the same duration of FU (6 months), the result was significant (MD 0.90 [0.32, 1.48], $I^2$ = 76%, *n* = 182), with reduced heterogeneity (S1 File).

**Mucociliary clearance.** Five studies reported MC as a percentage [78, 79, 81, 86, 87], and none showed baseline imbalance (S1 File). The pooled result of all five studies at EoT found that there was greater improvement in MC in the CHM groups (MD 7.71 [4.31, 11.12], $I^2$ = 87%, *n* = 486), with considerable heterogeneity (Fig 8).

A sensitivity analysis of two 4-week studies [78, 79] showed significant improvement in the IM group (MD 9.58 [7.51, 11.65], $I^2$ = 0%, *n* = 180), with no heterogeneity (S1 File). The pooled result of three studies of *Bi yuan tong qiao ke li* showed a significantly greater increase

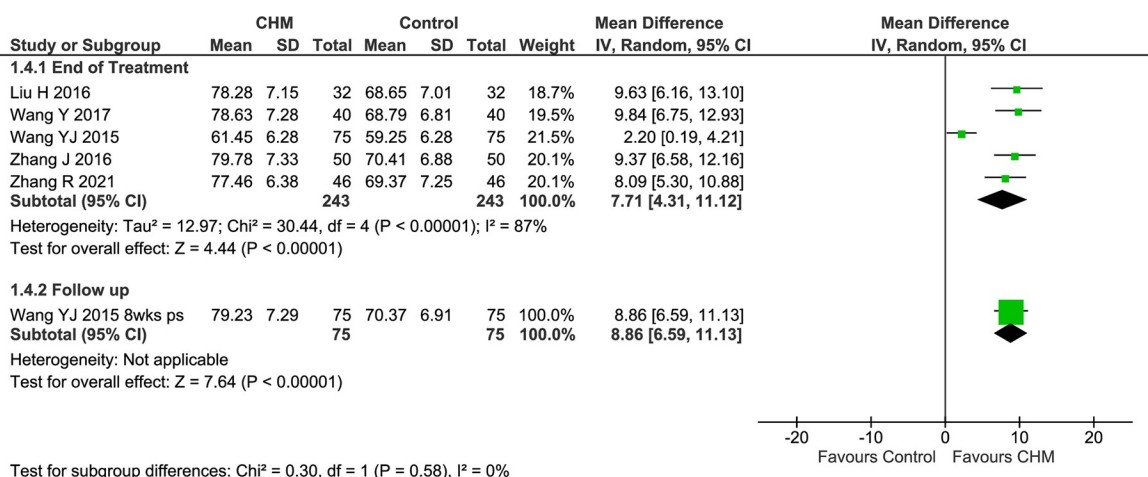

**Fig 8. Forest plot of CHM for CRS post-surgery: MC.** Abbreviations: CHM: Chinese herbal medicine; MC: Mucociliary clearance; ps: post-surgery; wks: weeks.

in MC (MD 7.04 [1.71, 12.38], $I^2$ = 92%, $n$ = 330), with considerable heterogeneity. In the three studies of CRSsNP, there was a significantly greater increase (MD 9.59 [7.82, 11.37], $I^2$ = 0%, $n$ = 244) with no heterogeneity. A similar result was evident when one of the studies was excluded so that all participants had the same CM syndrome and received the same CHM (MD 9.58 [7.51, 11.65], $I^2$ = 0%, $n$ = 180).

In the single study that reported follow-up data [87], there was still a significant improvement at 8 weeks post-surgery (MD 8.86 [6.59, 11.13]).

## Adverse events

Eleven studies did not mention of AEs, and seven reported there were no AEs in either group (S1 File). Three studies reported specific AEs [72, 73, 90], but all were minor. There were slightly fewer AEs in the groups that received CHMs compared with groups that only received conventional post-surgical management (12 versus 14), There were two dropouts in one study [80]—one in each group—but these were not related to the study medications. Overall, the AE data were insufficient for a complete safety analysis.

## GRADE assessments

The GRADE assessments (S1 File) were downgraded for each outcome by one GRADE due to the lack of blinding in the studies and by one GRADE for lack of a protocol and consequent risk of selective outcome reporting. For SNOT-20 (FU), LM (FU), MC (EoT), and MC (EoT, 4 weeks), the assessments were further downgraded for small sample size (less than 400) to very low certainty. For VAS-TNS (EoT), LK (EoT), LK (longest FU), MTT (EoT), MTR (EoT), and MTR (longest FU), the sample sizes were over 400 participants but there was significant heterogeneity in the meta-analysis result. Therefore, these were each downgraded to very low certainty. For SNOT-20 (EoT), LK (FU, 12 weeks post-surgery), and MTR (FU, 6 months post-surgery), each was downgraded to very low certainty.

Overall, there were significantly greater improvements at EoT in each outcome, except for SNOT-20, in the integrative medicine groups that received oral CHMs post-surgery. For the outcomes that had poolable data at FU (including SNOT-20), each showed a significantly greater improvement in the integrative medicine groups compared to the groups that had only

received usual post-surgical care. This suggested that the addition of the oral CHMs improved recovery after surgery for CRS; however, most of the studies were not blinded, which limited our certainty regarding this evidence.

## Discussion

In summary, there was considerable variation in the number of studies that reported each of the outcome measures, with VAS-TNS, LK and MTR providing the most data.

For SNOT-20, two studies reported data at EoT, but the results were conflicting. The pooled result ($n = 162$) showed no significant difference between groups, but the heterogeneity was considerable. However, at the six-month FU, the pooled result of two studies ($n = 147$) showed a significantly greater improvement in total SNOT-20 scores in the groups that received CHMs, with no heterogeneity. Future studies could consider using SNOT-8 as a more feasible and accessible subjective score for symptoms assessment [59].

Eight studies reported VAS-TNS at EoT, and the pooled result ($n = 576$) showed greater symptom reduction in the CHM groups, but the heterogeneity was considerable. Excluding studies that did not report baseline data resulted in a pooled result for five studies ($n = 404$) that still showed a significant difference with reduced heterogeneity.

The LM scores were only reported in two studies at FU ($n = 228$). The pooled result showed a greater improvement in the CHM groups without important heterogeneity. In contrast, the LK scores were reported by 10 studies at EoT ($n = 862$). The pooled result showed a greater improvement in the CHM groups with moderate heterogeneity, and the Funnel plot did not show signs of publication bias. At the longest FU (8 weeks–24 months post-surgery), there was a significantly greater improvement in LK in the CHM groups in six studies ($n = 531$), but the heterogeneity was considerable. When the FU time was the same (12 weeks post-surgery), a sensitivity analysis still showed significant benefits with reduced heterogeneity, but this was based on two studies only ($n = 254$).

The pooled result for MTT was based on five studies ($n = 486$) and found significantly reduced transport time at EoT in the groups that received CHMs, but the heterogeneity was considerable. For MTR, data were available at EoT from nine studies ($n = 778$). The pooled result showed a significantly greater improvement in the CHM groups at EoT, with considerable heterogeneity. The sensitivity analyses showed similar results, but the considerable heterogeneity remained. At longest FU (8 weeks – 6 months post-surgery), the pooled result of five studies ($n = 545$) also showed a greater improvement in MTR in the CHM groups, with considerable heterogeneity. When the FU time was the same (6 months post-surgery) the MTR result was similar with reduced heterogeneity, but only two studies ($n = 182$) could be included. For MC as a percentage, data were available from five studies ($n = 486$). The results showed a significantly greater increase in the CHM groups in the pooled results at EoT, but the heterogeneity was considerable. This was reduced to $I^2 = 0\%$ when only the two studies of four weeks duration ($n = 180$) were included, while the significant difference in MC remained.

In the pooled results for CRSwNP, there were significant differences between groups in favour of the CHM groups for VAS-TNS, LK, and MTR. For CRSsNP, there were significant differences in favour of the CHM groups for LK, MTT, and MTR, but not VAS-TNS. For the other outcomes, there were insufficient data.

From the perspective of Chinese medicine, CRS is caused by dampness heat (*shi re*) attack on the sinuses and deficiency of healthy *qi* (*zheng qi xu*). This study found that in addition to herbs commonly used for nasal diseases, such as *Magnolia biondii* (*xin yi*), *Angelica dahurica* (*bai zhi*) and *Xanthium sibiricum* (*cang er zi*), the CHM formulae frequently included herbs for clearing heat and removing dampness, such as *Scutellaria baicalensis* (*huang qin*);

strengthening the Spleen and removing dampness, such as *Poria cocos* (*fu ling*) and *Atracty-lodes macrocephala* (*bai zhu*); and/or tonifying the *qi*, such as *Astragalus membranaceus* (*huang qi*) and *Glycyrrhiza uralensis* (*gan cao*) (S1 File). This is consistent with Chinese medicine treatment based on syndrome differentiation [91].

From a clinical perspective, measures of symptoms (i.e., SNOT and VAS) and objective measures of the state of the surgical field, such as LM and LK, are probably the more important outcomes. Postoperative FU is mainly conducted through nasal endoscopy to observe the progress of intraoperative mucosal epithelialisation. In addition, recovery of sinus mucosal ciliary function can be tracked using measures of mucociliary transport.

The pooled results for LM and LK both showed greater improvements at 8–12 weeks FU, and one study still showed improvements at 24 months FU (S1 File). For MTR there were significant improvements at 12 weeks EoT and at six months FU (S1 File). These results suggested that integrative treatment with CHM was beneficial to the epithelialisation of the operative cavity and recovery of sinus mucosa function.

With regard to the best available evidence for specific CHM interventions, two of the three formulae that were tested in multiple studies were also included in clinical guidelines [91, 92]: *Shen ling bai zhu san* (SLBZS), modified (2 studies) and *Huang qin hua shi tang* (HQHST) (2 studies). Modified *Shen ling bai zhu san* was listed in a guideline for CRS with the syndrome Spleen *qi* deficiency (*pi qi xu ruo*) [91]. One of the two RCTs specified this syndrome as an inclusion criterion [82], while the other study [84] did not specify a syndrome. The pooled results showed improved symptoms based on VAS-TNS, with no heterogeneity (S1 File). *Huang qin hua shi tang* was listed in another guideline for CRS with the syndrome Spleen–Stomach dampness-heat (*pi wei shi re*) [92]. This syndrome was an inclusion criterion in both studies [88, 89]. The pooled results of these two RCTs showed improvements in VAS-TNS (S1 File) and the endoscopic profile based on LK (S1 File), but with considerable heterogeneity. It should be noted that these clinical guidelines were for CRS and did not include post-surgical recovery.

*Bi yuan tong qiao ke li* (BYTQK) (3 studies) is a commercial product for CRS. One textbook noted this formula was suitable for CRC with the syndrome Stagnant heat of Lung meridian (*fei jing yu re*) [93]. Of the three RCTs, one used syndrome differentiation based on Stagnant heat in the Gallbladder (*dan fu yu re*) [79]. The only poolable data were for measures of mucociliary clearance, which all showed significant improvements, but the heterogeneity was considerable (S1 File). Overall, the more clinically relevant results were for the first two formulae since these were included in clinical guidelines and reported significant improvements in symptoms (S1 File).

In comparison with the results of the systematic review of oral CHMs for RS [43], BYTQK was used in multiple studies included in both reviews. However, the two most frequent formulae from the earlier review, *Bi yuan shu kou fu ye* (BYSKFY) and its capsule form *Bi yuan shu jiao nang* (BYSJN), which were tested in four studies, and *Bi dou yan kou fu ye* (BDYKFY), which was tested in three studies, did not appear at all as interventions for CRS post-surgery. Future studies could consider testing these CHMs.

When the herbal ingredients were compared, the most frequent ingredients in the formulae from the previous review were the same as in this review: *Magnolia biondii* (*xin yi*), *Angelica dahurica* (*bai zhi*), *Xanthium sibiricum* (*cang er zi*), and *Scutellaria baicalensis* (*huang qin*). In experimental studies, extracts of each of these four plants have been reported to demonstrate anti-inflammatory, anti-allergic and antioxidant effects. In addition, antimicrobial effects have been reported for *Angelica dahurica*, *Xanthium sibiricum*, and *Scutellaria baicalensis*, while antiviral effects have been shown for extracts of *Xanthium sibiricum* and *Scutellaria baicalensis*

[76]. These pharmacological actions of the herbs included in multiple CHMs may provide insight in to how these CHMs exerted the clinical effects reported in the RCTs.

## Limitations

A major limitation with the results of these meta-analyses is the lack of blinding in all but one study. Therefore, participants would have known they were receiving an additional intervention, and this knowledge may have influenced their recovery or their self-reported outcomes. In addition, the investigators, hospital staff and clinicians would have known about the study, and this may have influenced their assessments. Conversely, the outcomes LM and LK may be considered as relatively objective and reliable, compared to self-reported VAS. In future studies, a placebo for the oral CHM should be used to enable blinding of participants and personnel. This would strengthen the validity and certainty of the evidence.

Other methodological issues included the lack of clear reporting of the randomisation method in 52% of studies. Poor randomisation may lead to bias in allocation and to differences between groups at baseline. In our analyses, we attempted to address this issue by undertaking baseline assessments. These showed no significant baseline differences for the main outcome measures, although there were some significant differences in VAS-IS (S1 File). However, some studies did not present baseline data, so these were excluded in the sensitivity analyses. While we can have more confidence in the pooled results of these sensitivity analyses, this method cannot completely mitigate any effects of poor randomisation practice.

Overall, heterogeneity marred many of the meta-analysis results. This was somewhat reduced in the sensitivity analyses. The additional sensitivity analyses for CRSwNP and CRSsNP reduced heterogeneity for LK and MC. While those for CM syndrome reduced heterogeneity for VAS-TNS, VAS-IS pain, LK, MTT and MC. However, this was at the expense of reducing the size of meta-analysis pools.

Since there were no study protocols available, it was likely that at least some of the studies did not report results for outcome measures that were not favourable to the test intervention. However, in the absence of a protocol we could not assess the prevalence of this as an issue. Future studies should be registered, and full protocols should be available to reviewers. The Consolidated Standards of Reporting Trials (CONSORT) [94, 95] should be followed to ensure rigorous, internationally- recognised methods are employed and all data are reported adequately.

There was a lack of safety reporting in most studies. Under-reporting of adverse drug reactions and AEs is a major issue internationally [96]. Pharmacovigilance largely depends on adequate reporting, so this is an important responsibility of clinicians and researchers to fully report all AEs according to established guidelines [64, 97, 98].

Although many CHMs contained common ingredients, there was considerable variation in the test interventions, with only three CHM formulae (BYTQK, HQHST and SLBZS) being tested in multiple (2 or 3) studies. This diversity in the CHMs was a likely source of heterogeneity in the meta-analyses. More high-quality RCTs are needed for previously tested CHMs to provide rigorous assessments of their effects.

With regards to the CHMs used in the studies, information on quality control was absent in the study reports. This is an important issue and needs to be a standard item in reports of clinical trials of CHMs [99–103].

Most meta-analysis pools showed heterogeneity which limited our certainty in the effect size estimates. This was reflected in the GRADE assessments, which were all assessed as very low certainty. The sources of the heterogeneity were difficult to determine despite the use of sensitivity analyses, but likely factors include real differences in the effects of specific herbal

medicines; variations in the severity of CRS; differences in the CRS endotypes the participants [104]; variability in surgical procedures and post-surgical care; differences in patient care following discharge; and possibly unreported baseline imbalances. However, it was not possible to determine the relative contributions of these factors. When all these details are provided in the study protocol and the study report, it may be possible to assess the factors that contribute to early recovery.

The relatively small sizes of the clinical studies and the meta-analysis pools (less than 500 participants) was a concern and led to down-grading of the evidence. Further studies with larger sample sizes are needed. Another limitation was that relatively few studies reported FU data, and only one study reported long-term FUs. It is important to determine whether any benefits associated with the addition of CHMs to usual care are sustained long term and whether there are any effects on rates of recurrence. Future studies should report FUs of 6–12 months.

Following further research, CHMs could become important integrative treatment options for CRS, especially for patients who do not achieve complete control of their symptoms with surgery and conventional medical management alone. A personalized, patient-centred approach that addresses the root causes of CRS from a both conventional and traditional medicine perspectives could assist in the management of difficult to treat conditions involving cephalea and rhinogenic obstruction [22].

## Conclusions

The meta-analyses suggest that certain CHMs may provide additional benefits in the post-surgical management of CRS, and these benefits may continue longer term as reductions in symptoms and improved recovery of the surgical field. The strengths of this review included comprehensive searches, analysis of recognised outcome measures, assessments of baseline balance, and inclusion of sensitivity analyses to improve effect size estimates. Weaknesses include the lack of blinding in most clinical studies, small sample sizes, incomplete reporting of baseline data and AEs, and few long-term FUs. Future studies should employ rigorous methodology, include a placebo for the CHM in the control group to ensure blinding, have adequate sample sizes, include quality control for the CHMs, include information on the syndrome and endotype of CRS, report all AEs and provide a detailed protocol.

## Supporting information

**S1 Checklist. PRISMA 2009 checklist.**
(DOCX)

**S1 File. Additional data.** Including: Funnel plot of results for LK at EoT; Databases that were searched and PubMed search terms for CHM for RS; Ingredients of the CHM interventions, syndrome, manufacturer, and funding sources; Main ingredients of the Chinese herbal interventions: All studies; Main ingredients of the Chinese herbal interventions: CRSwNP studies; Main ingredients of the Chinese herbal interventions: CRSsNP studies; Risk of bias judgements for included studies; SNOT 20: Meta-analysis results for chronic rhinosinusitis post-surgery; VAS-TNS: Meta-analysis results for chronic rhinosinusitis post-surgery; VAS-IS: Meta-analysis results for chronic rhinosinusitis post-surgery; LM: Meta-analysis results for chronic rhinosinusitis post-surgery; LK: Meta-analysis results for chronic rhinosinusitis post-surgery; MTT: Meta-analysis results for chronic rhinosinusitis post-surgery; MTR: Meta-analysis results for chronic rhinosinusitis post-surgery; MC: Meta-analysis results for chronic rhinosinusitis post-surgery; Details of reported adverse events from included studies; GRADE

assessments for each outcome measure.
(DOC)

## Acknowledgments

We wish to thank Dr Iris WY Zhou and Dr Meaghan Coyle for their assistance with searches, and Dr Meaghan Coyle for her comments and editing.

## Author Contributions

**Conceptualization:** Jing Cui, Wenmin Lin, Brian H. May, Christopher Worsnop, Anthony Lin Zhang, Xinfeng Guo, Chuanjian Lu, Charlie C. Xue.

**Data curation:** Jing Cui, Brian H. May, Qiulan Luo.

**Formal analysis:** Jing Cui, Wenmin Lin, Brian H. May, Qiulan Luo.

**Funding acquisition:** Chuanjian Lu, Charlie C. Xue.

**Investigation:** Jing Cui, Wenmin Lin, Brian H. May, Qiulan Luo, Anthony Lin Zhang.

**Methodology:** Wenmin Lin, Brian H. May, Anthony Lin Zhang, Xinfeng Guo, Chuanjian Lu.

**Project administration:** Anthony Lin Zhang, Xinfeng Guo, Chuanjian Lu, Charlie C. Xue.

**Resources:** Anthony Lin Zhang, Xinfeng Guo, Yunying Li.

**Supervision:** Brian H. May, Christopher Worsnop, Anthony Lin Zhang, Xinfeng Guo, Chuanjian Lu, Yunying Li, Charlie C. Xue.

**Writing – original draft:** Jing Cui, Wenmin Lin, Brian H. May.

**Writing – review & editing:** Brian H. May, Qiulan Luo, Christopher Worsnop, Anthony Lin Zhang, Xinfeng Guo, Chuanjian Lu, Yunying Li, Charlie C. Xue.

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
