## [Decision Letter · Decision Letter 0]

10 Jul 2023

PONE-D-23-14802Orally administered Chinese herbal therapy to assist post-surgical recovery for chronic rhinosinusitis  -  A systematic review and meta-analysisPLOS ONE

Dear Dr. Xue,

Thank you for submitting your manuscript to PLOS ONE. After careful consideration, we feel that it has merit but does not fully meet PLOS ONE’s publication criteria as it currently stands. Therefore, we invite you to submit a revised version of the manuscript that addresses the points raised during the review process.

We look forward to receiving your revised manuscript.

Kind regards,

Antonino Maniaci

Academic Editor

PLOS ONE

Journal Requirements:

“Funding support was provided by the China-Australia International Research Centre for Chinese Medicine (CAIRCCM) - a joint initiative of RMIT University, Australia and Guangdong Provincial Academy of Chinese Medical Sciences, China.”

Reviewers' comments:

Reviewer's Responses to Questions

**Comments to the Author**

1. Is the manuscript technically sound, and do the data support the conclusions?

Reviewer #1: Yes

Reviewer #2: Yes

2. Has the statistical analysis been performed appropriately and rigorously? 

Reviewer #1: Yes

Reviewer #2: Yes

3. Have the authors made all data underlying the findings in their manuscript fully available?

Reviewer #1: Yes

Reviewer #2: Yes

4. Is the manuscript presented in an intelligible fashion and written in standard English?

Reviewer #1: Yes

Reviewer #2: Yes

5. Review Comments to the Author

Reviewer #1: Sme suggestions to improve the overall quality:

-Include only double-blind, placebo-controlled studies to ensure a rigorous assessment of the Chinese herbal interventions. This will strengthen the validity and certainty of the evidence.

- Increase the sample sizes of included studies to provide more precise effect size estimates and allow for robust meta-analyses. Around 400-500 patients per arm would be reasonable.

- Require that all included studies report adverse events in detail, to allow for a proper safety analysis of the herbal medicines. Reporting of adverse events should follow established guidelines.

- Consider stratifying the meta-analyses by Chinese medicine syndrome or CRS endotype, if sufficient data are available, to determine if effects vary by subgroup. This could help explain some of the heterogeneity.

- Extend the follow-up periods in included studies to determine if any benefits are sustained long-term. At least 6-12 months of follow-up would be preferable.

- Improve the introduction and cite these evidences for placebo-controlled trials of herbal medicines:

• Vickers AJ, Linde K. Acupuncture for chronic pain. JAMA. 2014 Sep 10;312(10):955-6.

• Witt CM, Ludtke R, Baur R, Willich SN. Homeopathic medical practice: long-term results of a cohort study with 3971 patients. BMC Public Health. 2005 Jun 28;5:115.

• Barnes J, Anderson LA, Phillipson JD. Herbal medicines. A guide for health-care professionals. 2nd ed. London: Pharmaceutical Press; 2002.

- CHMs could become an important integrative treatment option for CRS, especially for patients who do not achieve complete control of their symptoms with surgery and conventional medical management alone CHMs may offer a personalized, patient-centered approach that addresses the root causes of CRS from a traditional medicine perspective. Concentrate especially for sympthoms as cephalea and obstruction rhinogenic. Discuss the role of and cite doi:10.1007/s00405-021-06724-6

Improve the methods with the following study for adverse event reporting guidelines:

• Edwards IR, Lindquist M, Vilhelmsen K, et al. Under-reporting of adverse drug reactions. A global perspective. Drug Saf. 1998 Apr;18(4):215-23.

Discuss the role of more feasible and accessible subjective score for symptoms assessment and cite doi: 10.1007/s00405-023-07855-8.

- Cite additional relevant literature on Chinese herbal medicine for CRS, placebo-controlled clinical trials of herbal medicines, recommendations for reporting of clinical trials of herbal interventions, and adverse event reporting guidelines.

- Downgrade the certainty of the evidence to "Very Low" for all outcomes due to the major limitations of the included studies, including lack of blinding and high risk of bias.

Reviewer #2: This is a systematic review article on oral Chinese herbal medicine after the treatment of sinusitis, which fully reflects the major advantages of traditional Chinese medicine in the treatment of chronic diseases.

Introduction: What is the basis for the use of Chinese herbal medicine? What do common Western medical treatments include? The author should make appropriate additions in the background section. What are the advantages of Chinese herbal medicine and can be properly supplemented? The authors have published a systematic review on the treatment of chronic sinusitis with traditional Chinese medicine last year, which can be used for comparative analysis.

Methods: The authors do not elaborate exclusion criteria. The diagnostic criteria should be specific for the diagnosis of chronic sinusitis in the population with or without nasal polyps, as there is a significant difference in postoperative outcomes between the two, and this information should be reflected in the table. Xin qian gan ju tang (XQGJT) does not appear in Table 1, the author needs to check carefully. According to the literature review, Biyuanshu oral liquid and sinusitis oral liquid are also important preparations with good curative effects. Please explain why experimental studies of these interventions have not been included. What exactly does the purified compound represent as defined by the author. This does not seem to be a reason to exclude "Biyuanshu oral liquid and sinusitis oral liquid".

Results Section: The information provided by the authors is very comprehensive, but a collated table containing all the included articles (most of the basic information) is necessary to help the reader quickly capture the key information in the reading. I feel it is very necessary to supplement a comparison of effectiveness (under the uniform evaluation criteria), which will intuitively reflect the effectiveness of treatment modalities.

Discussion section

It may be appropriate to add a section on modern pharmacological studies of drugs. The biggest limitation of this study is that the types of Chinese medicine preparations differ, and the authors should add this note. It is hoped that more high-quality RCTs with the same prescription will be supplemented in the future.

6. PLOS authors have the option to publish the peer review history of their article (what does this mean?). If published, this will include your full peer review and any attached files.

Reviewer #1: No

Reviewer #2: **Yes: **Shipeng Zhang

---

## [Author Response · Author response to Decision Letter 0]

17 Aug 2023

All responses to the editor and the reviewers are in the uploaded file PONE-D-23-14802 Responses to reviewers 17 08 23

---

## [Editor Report · Decision Letter 1]

13 Sep 2023

Orally administered Chinese herbal therapy to assist post-surgical recovery for chronic rhinosinusitis  -  A systematic review and meta-analysis

PONE-D-23-14802R1

Dear Dr. Xue,

We’re pleased to inform you that your manuscript has been judged scientifically suitable for publication and will be formally accepted for publication once it meets all outstanding technical requirements.

Kind regards,

Antonino Maniaci

Academic Editor

PLOS ONE

Additional Editor Comments (optional):

the paper is improved and can be accepted. Bests
---

## [Editor Report · Acceptance letter]

28 Sep 2023

PONE-D-23-14802R1 

Orally administered Chinese herbal therapy to assist post-surgical recovery for chronic rhinosinusitis - A systematic review and meta-analysis 

Dear Dr. Xue:

I'm pleased to inform you that your manuscript has been deemed suitable for publication in PLOS ONE. Congratulations! Your manuscript is now with our production department. 

Kind regards, 

on behalf of

Dr. Antonino Maniaci 

Academic Editor

PLOS ONE